# The Role of Muscarinic Acetylcholine Receptor M_3_ in Cardiovascular Diseases

**DOI:** 10.3390/ijms25147560

**Published:** 2024-07-10

**Authors:** Xinxing Liu, Yi Yu, Haiying Zhang, Min Zhang, Yan Liu

**Affiliations:** 1Hainan Academy of Medical Sciences, School of Pharmacy, Hainan Medical University, Haikou 571199, China; liuxinxing56@163.com (X.L.); yewiran@163.com (Y.Y.); hyzhang@hainmc.edu.cn (H.Z.); 2Engineering Research Center of Tropical Medicine Innovation and Transformation of Ministry of Education, Hainan Academy of Medical Sciences, Hainan Medical University, Haikou 571199, China; 3International Joint Research Center of Human–Machine Intelligent Collaborative for Tumor Precision Diagnosis and Treatment of Hainan Province, Hainan Academy of Medical Sciences, Hainan Medical University, Haikou 571199, China; 4Hainan Provincial Key Laboratory of Research and Development on Tropical Herbs, Hainan Academy of Medical Sciences, Hainan Medical University, Haikou 571199, China

**Keywords:** muscarinic acetylcholine receptor M_3_, cardiomyocyte, endothelial cell, fibroblast, cardiovascular diseases

## Abstract

The muscarinic acetylcholine receptor M_3_ (M_3_-mAChR) is involved in various physiological and pathological processes. Owing to specific cardioprotective effects, M_3_-mAChR is an ideal diagnostic and therapeutic biomarker for cardiovascular diseases (CVDs). Growing evidence has linked M_3_-mAChR to the development of multiple CVDs, in which it plays a role in cardiac protection such as anti-arrhythmia, anti-hypertrophy, and anti-fibrosis. This review summarizes M_3_-mAChR’s expression patterns, functions, and underlying mechanisms of action in CVDs, especially in ischemia/reperfusion injury, cardiac hypertrophy, and heart failure, opening up a new research direction for the treatment of CVDs.

## 1. Introduction

Muscarinic acetylcholine receptor M_3_ (M_3_-mAChR) is a member of the superfamily of G protein-coupled receptors, which are characterized by seven transmembrane domains [1] and expressed by cardiomyocytes [2,3,4], stem cells [5], beta-cells [6,7], neurons [8,9], smooth muscle cells [10,11,12], and a variety of epithelia [13,14,15]. Choline binds to M_3_-mAChR to regulate a variety of pathophysiological processes, including cardiac injury [16,17], cardiac senescence [18], cardiac fibrosis [19], cardiac hypertrophy (CH) [20,21], immune regulation [22,23,24], and metabolic regulation [25,26].

Cardiovascular diseases (CVDs) are the leading cause of mortality globally, accounting for about 17.9 million deaths in 2019 [27]. Strikingly, ~85% of CVD-related deaths are due to heart attack and stroke. CVDs include ischemia/reperfusion (I/R) injury, arrhythmias, myocardial infarction, atherosclerosis, cardiac senescence, cardiac fibrosis, CH, and heart failure (HF). Multiple risk factors (such as hypertension, diabetes mellitus, obesity, and aging) and the interconnected networks of various molecular signaling pathways involved in disease onset and progression have hindered the progress in treatment strategies for CVDs. Thus, continued efforts are needed to identify additional therapeutic targets.

The activation of M_3_-mAChR exerts cardioprotective effects on CVDs through the NF-κB/miR-376b-5p/BDNF axis [3], inhibition of vascular remodeling by the activation of the nuclear factor erythroid 2-related factor 2 pathway [10], delay of cardiac senescence by the inhibition of the caspase-1/interleukin (IL)-1β signaling pathway [2], and the suppression of ischemia-induced arrhythmia by reducing the Ca^2+^ overload [28].

The aim of this review article is to summarize the various cardioprotective effects produced after the activation of M_3_-mAChR and provide a clear description of M_3_-mAChR’s protein structure, expression patterns, functions, and underlying mechanisms of action in CVDs, especially in I/R injury, CH, and HF.

## 2. Muscarinic Acetylcholine Receptors

As first described by Loewi and Dale in 1921 [29], human mAChRs are classified into five subtypes (M_1_–M_5_), which are encoded by the genes *CHRM_1_* to *CHRM_5_* and differ by preference in G-protein coupling. Subtypes M_1_, M_3_, and M_5_ couple to G proteins of the G_q/11_ family and activate phospholipase C, while subtypes M_2_ and M_4_ couple to G_i/o_-type G proteins and inhibit adenylyl cyclase and modulate ion channel functions [1,30,31,32] (Figure 1).

In addition, they are widely distributed throughout the body and maintain peripheral autonomic functions and central nervous system control of arousal, attention, memory, and motivation [33]. M_1_-mAChR has been found in the brain, glands, small intestine, heart muscle, kidney, lung, skin, stomach, and smooth muscle [34], playing an essential role as a vasoconstrictor to increase heart rate (HR) [35,36], contractile force [37], and diastolic function [38,39]. M_2_-mAChR has been identified in the brain, heart muscle, colon, liver, lung, placenta, small intestine, smooth muscle, and urinary bladder [34], and it modulates pacemaker activity, atrioventricular conduction, contractility, and sympathetic neurotransmitter release in the atria, and promotes vasodilation [40,41]. M_3_-mAChR, which exists in the brain, glands, stomach, smooth muscle, spleen, small intestine, lung, kidney, and heart muscle [34], is involved in acetylcholine (ACh)-induced endothelium-dependent dilation of the coronary arteries [42], cell–cell communication [43], regulation of HR and repolarization [44], activation of survival pathways [45], and vasoconstriction and relaxation [46,47]. M_4_-mAChR, which has been found in the spleen, brain, duodenum, small intestine, and testis [34], plays important roles in sympathetic neurotransmitter release in the atria [40] and in the regulation of K^+^ channels [44,48]. M_5_-mAChR is expressed in the brain, placenta, and testis [34]. Overall, mAChRs regulate a very large number of important pathophysiology and physiological processes, especially selective cardiac protection.

## 3. Choline and M_3_-mAChR

Choline is the precursor of the vagal neurotransmitter ACh and acts as an agonist of mAChR. Choline functions in the neurological system via the activation of cholinergic receptors, including M_3_-mAChR. Additionally, choline is essential for the formation and integrity of cell membranes, lipid metabolism, and synthesis of phospholipids [49]. Choline, via the activation of M_3_-mAChR, protects against arrhythmia [28], CH [21], ischemic myocardial injury [45], reperfusion injury [50], vascular remodeling [10], myocardial fibrosis [19,51], and myocardial infarction [52].

Choline has a concentration-dependent impact that can lead to a decrease in action potentials’ (APs) length, a reduction in sinus rhythm, and activation of a K^+^ current [53]. In addition, choline was reported to exhibit anti-arrhythmic effects by decreasing the density of L-type Ca^2+^ currents in cardiac cells in verapamil- or aconitine-induced arrhythmias [54]. Choline also protects endothelial cells (ECs) against the damage caused by excessive glucose, endoplasmic reticulum (ER) stress, and apoptosis during hypoxia/reoxygenation (H/R) [55]. Via the activation of M_3_-mAChR, choline can decrease angiotensin II (Ang II)-induced apoptosis of cardiomyocytes [56], maintain intracellular levels of calcium and reactive oxygen species (ROS), reverse the effects of Ang II on atrial natriuretic peptide and beta-myosin heavy chain levels in cardiomyocytes, and reduce upregulated levels of phosphorylated p38 mitogen-activated protein kinase (p38MAPK) and calcium/calmodulin kinase II (CaMKII) in Ang II-induced CH [21].

Furthermore, choline was found to shield H9c2 cells against etoposide-induced apoptosis via the activation of M_3_-mAChR. Notably, this effect was amplified via the overexpression of M_3_-mAChR. In H9c2 cells, the protective effects of M_3_-mAChR are elicited by heme oxygenase-1 (HO-1)-induced upregulation of hypoxia-inducing factor 1 alpha (HIF-1α) and vascular endothelial growth factor (VEGF) [57]. These findings imply that M_3_-mAChR activation is necessary for choline to exert cardioprotective benefits.

## 4. Acetylcholine and M_2_/M_3_-mAChR

ACh is a key protective molecule in heart disease, which acts through the parasympathetic branches of the autonomic nervous system and sympathetic nervous system [58], as well as in isoallol-induced CH [59,60], acute myocardial infarction [61], Ang II-induced cardiac dysfunction [62], and other conditions that can exert cardiovascular protection.

In the heart, the M_2_-mAChR type is the predominant isoform and is abundantly expressed in the atrium. M_3_-mAChR is also expressed in cardiomyocytes but is much less abundant than M_2_-mAChR [63,64]. Existing studies have shown that ACh release in the human atrium is completely controlled by M_2_-mAChR [65]. ACh slows HR by decreasing sinoatrial node firing rate and atrioventricular node conduction velocity by activating atrial M_2_-mAChR subtypes, resulting in negative inotropy; ACh-activated M_2_-mAChR also attenuates β_2_-Adrenergic receptor and α_1_-Adrenergic receptor signaling [66]. At present, many studies have described the role of M_2_-mAChR in the heart, but there are still few relevant studies on the protective effect of M_3_-mAChR on the heart [64].

In rat cardiomyocytes mimicking I/R injury, ACh protected H9c2 cells from H/R-induced oxidative stress via M_2_-mAChR in a concentration-dependent manner [67]. ACh is also able to stimulate cytoprotective mitophagy by promoting PTEN-induced kinase 1/Parkin mitochondrial translocation via the M_2_-mAChR, which provides a beneficial target for protecting the cardiac internal environment from H/R injury [68]. In patients with dementia, ACh has a protective effect on cardiovascular outcomes, especially in reducing HF hospitalization and myocardial revascularization [69]. Existing research is gradually revealing the effect of ACh on M_3_-mAChR. In ECs, ACh-activated M_3_-mAChR inhibits H/R-induced ER stress and apoptosis through the AMPK signaling pathway [55,70].

## 5. M_3_-mAChR in Different Cardiac Cell Types

The adult human heart is primarily composed of cardiomyocytes, fibroblasts, and ECs [71], which all express M_3_-mAChR [72]. M_3_-mAChR participates in the regulation and maintenance of cardiac function, which depends on the cell type [73]; hence, clarification of the function of M_3_-mAChR in different cardiac cell types is crucial for the effective treatment of CVDs.

### 5.1. M_3_-mAChR in Cardiomyocytes

M_3_-mAChR plays vital roles in cardiomyocyte function, metabolism, senescence, and apoptosis. However, additional studies are needed to further clarify the role of M_3_-mAChR in cardiac physiology and pathophysiology, and to provide new strategies for the prevention, diagnosis, and management of CVDs.

#### M_3_-mAChR Signal in Cardiomyocytes

In support of the precious of M_3_-mAChR in cardiomyocytes, prior studies have demonstrated that muscarinic receptors are coupled with K^+^ channels in the heart and that a delayed rectifying potassium current triggered by M_3_-mAChR stimulation, known as I_KM3_, exhibits a linear relationship between current and voltage [72,74,75]. Increased repolarizing K^+^ currents stop harmful electrical remodeling in CH. In mouse central ventricular cardiomyocytes, overexpression of M_3_-mAChR specifically enhanced I_K1_, I_to_, and K_ir2_._1_ current densities [76]. The balance between Ca^2+^ and K^+^ currents in cardiomyocytes, which prolong and promote repolarization, respectively, determines the duration of APs. Activated M_3_-mAChR can mediate K^+^ currents. For example, in the working myocardium and cardiac pacemakers of mice, M_3_-mAChR causes atrial myocytes to produce outward K^+^ currents to accelerate membrane hyperpolarization, shorten AP duration, and reduce the rate of depolarization, thereby lowering HR and the sinus rhythm to protect against cardiac arrhythmia [28,72]. These investigations provide evidence that M_3_-mAChR is involved in the regulation and maintenance of heart function.

Cardiac dysfunction alters the circadian rhythms of clock genes and has been linked to the overexpression of proteins involved in Ca^2+^ entry into cardiomyocytes. Choline can ameliorate the disruption of circadian rhythms, lower the amounts of proteins that handle Ca^2+^, and improve cardiac remodeling and dysfunction. Additionally, choline improves Ca^2+^ overload by reducing entry in cardiomyocytes [77]. Furthermore, choline protects mitochondrial function and reduces the increased mitochondrial production of ROS to prevent CH. Moreover, choline suppresses metabolic dysfunction and stimulates the production of proteins involved in fatty acid metabolism through the SIRT3-AMPK signaling pathway in the heart [49]. M_3_-mAChR inhibitors block the cardioprotective effects of choline, suggesting that M_3_-mAChR activation underlies the cardioprotective effects of choline. In addition, activated M_3_-mAChR reduces the size of cardiomyocytes and inhibits apoptosis, thereby further alleviating cardiac remodeling [56]. Choline activates M_3_-mAChR can inhibit the activation of the P38MAPK signaling pathway and downregulate the expression of the Ang II type 1 receptor [20] against CH, and it exhibits anti-apoptotic activities via the HO-1-induced upregulation of HIF-1α and VEGF [57]. In cardiomyocytes, miR-376b-5p promotes myocardial ischemic injury by inhibiting BDNF; M_3_-mAChR mediates cardiac protection by downregulating miR-376b-5p through NF-κB [3].

In a mouse model of D-galactose-induced cardiac aging, M_3_-mAChR expression was downregulated, and M_3_-mAChR activation suppressed the increased production of IL-1β and caspase-1 in cardiomyocytes, thus delaying aging [2]. In the normal rat cardiomyocytes, activated M_3_-mAChR was shown to block L-type Ca^2+^ channels, which then accelerated the repolarization of APs. However, in the aged heart, downregulated M_3_-mAChR expression could not control electrical activity [72,78]. Additionally, M_3_-mAChR contributes to intercellular communication via gap junction proteins, and the rate of repolarization in cardiomyocytes may be regulated by M_3_-mAChR and the gap junction protein Cx43 [4].

In summary, M_3_-mAChR controls the balance of Ca^2+^ and K^+^ currents in cardiomyocytes and, ultimately, the electrical activity of the heart. Furthermore, M_3_-mAChR actively takes part in cardiomyocyte senescence and apoptosis. As a cardioprotective effect, activated M_3_-mAChR inhibits the apoptotic pathway.

### 5.2. M_3_-mAChR in Endothelial Cells

ECs, as one of the most significant components of the cardiovascular system, are dynamic regulators of vascular tone and local cell proliferation [79]. ECs guide and promote the growth of cardiomyocytes, diastolic function, survival following ischemia injury, and even provide limited regeneration by supplying blood oxygen and nutrients, in addition to autocrine and paracrine signaling.

#### M_3_-mAChR Signal in Endothelial Cells

M_3_-mAChR has been shown to impact vascular ECs and, consequently, cardiac function. ACh can inhibit ER stress and apoptosis in ECs induced by H/R by activating the AMPK signaling pathway through M_3_-mAChR; this protective effect can be abolished by 4-diphenylacetoxyl-N-methylpiperidine (4-DAMP) [55]. In an animal model fed with a high-fat diet, pyridostigmine increased serum ACh levels and activated M_3_-mAChR in HUVECs, which reduced the markers of ER stress, O-glycosylation, and apoptosis [70]. In addition to aiding in the synthesis of adenosine triphosphate (ATP), the mitochondria of vascular ECs help to maintain normal physiological levels of Ca^2+^, ROS, and nitric oxide. The cardioprotective effects of M_3_-mAChR also involve the inhibition of both the H/R-induced mitochondrial unfolded protein response, which maintains mitochondrial morphology and function, and Ca^2+^ excess in the mitochondria [80]. ACh inhibits H/R through M_3_-mAChR-mediated unfolded protein response UPR (mt) in vascular ECs, resulting in beneficial effects [81].

M_3_-mAChR is linked to the function of the endothelial barrier; thus, a knockdown greatly increased the permeability of the vascular endothelium and the effect of increased 4-DAMP concentrations on the permeability of ECs [82].

Notably, M_3_-mAChR induced endothelium-dependent diastole of the small arteries in vivo [15]. However, the endothelial depletion of M_3_-mAChR has implications for the cardiovascular system. Therefore, the physiological role of M_3_-mAChR in ECs remains unclear. Existing studies on M_3_-mAChR expression in ECs are still insufficient, and further studies are needed to explore this aspect.

### 5.3. M_3_-mAChR in Smooth Muscle Cells

Vascular smooth muscle cells (SMCs) are highly specialized cells whose main function is to regulate the structural integrity, vascular tone, and blood flow distribution of blood vessels. Mature SMCs proliferate at a very low rate, have a very low synthetic activity, and express a range of cell contract-related proteins, ion channels, and signaling molecules [83]. SMCs have strong plasticity and play different roles under different conditions. For example, in the process of vascular development, SMCs play a key role in vascular morphology, significantly improve the rate of cell proliferation, migration, and synthesis when dealing with vascular injury, and play a key role in vascular repair. Behind this high plasticity, they may contribute to the development of vascular diseases [84].

#### M_3_-mAChR Signal in Smooth Muscle Cells

Cardiac protein-related transcription factors [myocardin family of transcriptional coactivators (MRTFs): myocardin, MKL-1/MRTF-A, and MKL-2/MRTF-B] and the serum response factor play an important role in SMCs [85]. Using RT-qPCR confirmed that M_3_-mAChR expression was elevated in SMCs after MRTF-B treatment, and MRTFs were effective for M_3_-mAChR. However, MRTF-B was the most powerful transactivator of M_3_-mAChR, a finding that provides a new transcriptional control mechanism that can be used for clinical treatment [86]. Methacholine induces an enhancement in the phosphoinositide metabolism within bovine tracheal smooth muscle. M_3_-mAChR in the heart mediates phosphatidylinositol turnover [87]. The presence of M_3_-mAChR has been identified in most SMCs [88], and the presence of M_3_-mAChR is critical for gastric smooth muscle, enabling it to contract [89,90].

At present, there are few studies on the role of M_3_-mAChR in SMCs, and the research focuses on the gastrointestinal tract. Therefore, it is worth exploring the role of M_3_-mAChR in the smooth muscle within the heart.

### 5.4. M_3_-mAChR in Fibroblasts

Apart from the cells that make up the myocardium and vascular system, fibroblasts are the most common type of cardiac cells, accounting for ~20% of the mouse heart, 24.3% of the human atria, and 15.5% of the human ventricles [91]. In addition, fibroblasts are crucial for the development of cardiac fibrosis, various other cardiac disorders, and for the development and physiological functions of the myocardium [92]. In the heart, fibroblasts control collagen renewal. Dysregulation of collagen synthesis, metabolism, and degradation can result in severe structural damage, systolic and diastolic abnormalities, and fibrosis of cardiac tissues. Myofibroblasts function as the principal effector cells of collagen synthesis and the primary mediators of fibrosis [93].

#### M_3_-mAChR Signal in Fibroblasts

Choline was reported to mitigate the effects of adriamycin against oxidative stress, inflammation, and apoptosis [94]. Choline also restores the balance of the autonomic nervous system, mitigates decreased cardiac function, and ameliorates myocardial fibrosis by increasing ACh levels and the high-frequency component of HR variability while decreasing norepinephrine levels and the low-frequency component of HR variability. Choline also upregulates the antioxidant markers’ nuclear factor erythroid 2-related factor 2 and HO-1. Simultaneously, the muscarinic receptor antagonist atropine partially counteracts the protective effect of choline, indicating that choline functions via vagal activation. These findings suggest that choline could be useful as an adjuvant drug [95].

In addition, choline was found to inhibit the progression of cardiac fibrosis via the activation of M_3_-mAChR [45,51], which resulted in a decreased production of collagen I and III in a mouse model of aortic constriction and cell pretreats with TGF-β1 [19,96]. These findings confirm that choline-activated M_3_-mAChR is expressed in fibroblasts and prevents TGF-β1-induced fibroblast formation to prevent cardiac fibrosis through TGF-β1/Smad2/3 and p38MAPK pathways. Existing studies are still unable to directly demonstrate the physiological effects and expression changes of M_3_-mAChR in fibroblasts, which deserve our detailed consideration.

## 6. M_3_-mAChR and Cardiovascular Diseases

CVDs, including ischemic heart disease, stroke, HF, and peripheral arterial disease, are the most common causes of death globally and are the major contributors to a decreased quality of life. M_3_-mAChR is involved in the regulation and pathophysiology of CVDs (Figure 2) (Table 1).

### 6.1. M_3_-mAChR in Ischemia/Reperfusion

Typically, coronary atherosclerosis causes severe and prolonged myocardial ischemia, resulting in irreparable myocardial infarction, possibly due to reduced blood flow through narrowed coronary arteries to the myocardium [97]. Prompt blood reperfusion could prevent myocardial infarction but also cause irreversible I/R injury. Since there is currently no approved medication for I/R injury, further studies are needed to identify potential therapeutic targets [98].

A cardioprotective signaling cascade against I/R injury can be triggered via the activation of M_3_-mAChR. The activation of M_3_-mAChR by choline can significantly reduce the infarct size and exert cytoprotection on ischemic myocardium. MAChR participates in the anesthetic preconditioning induced by I/R in rats in vitro cardiac cholinergic receptor activation. Moreover, the process of cholinergic receptor-mediated by the activation of eNOS phosphorylation, nNOS, and CaMKK β/AMPK plays a role of myocardial protection [99]. Choline was shown to decrease ROS levels and vascular dysfunction, prevent the overexpression of phosphorylated and oxidized CaMKII, and correct abnormal expression of Ca^2+^-cycling proteins, including receptor phosphoproteins, inositol trisphosphate receptor, sarcoplasmic reticulum Ca^2+^-ATPase, and Na^+^/Ca^2+^ exchange proteins. The vascular protective effect of choline was mediated by the stimulation of M_3_-mAChR. This may be a new therapeutic strategy for the treatment of I/R-induced vascular injury [50].

Increased autophagy and cardiomyocyte apoptosis have been linked to cardiac dysfunction during myocardial I/R. Choline suppresses the autophagic activity generated by myocardial I/R via the activation of the Akt/mTOR signaling pathway while suppressing autophagosome accumulation and cardiac apoptosis during I/R [100]. Choline has a protective effect on myocardial I/R by inhibiting excessive autophagy, and whether it plays a role in activating M_3_-mAChR remains to be studied. This provides a good entry point to further study whether M_3_-mAChR is involved in myocardial autophagy in I/R injury.

In isolated cardiomyocytes and a rat model of myocardial infarction, choline was found to mitigate myocardial injury by reducing ventricular arrhythmias, correcting hemodynamic deficits, and shielding cardiomyocytes from apoptotic cell death. These effects were countered by 4-DAMP but not M_2_-selective antagonists, indicating a significant protective role of M_3_-mAChR. Choline enhanced heart function and reduced ischemic myocardial damage via the stimulation of cardiac M_3_-mAChR and related signaling pathways [45].

Thus, M_3_-mAChR could be a promising drug target to enhance cardiac performance during I/R.

### 6.2. M_3_-mAChR in Cardiac Hypertrophy

The term hypertrophy refers to enlargement of the myocardium and individual cardiomyocytes to increase contractility of the heart under various conditions by increasing monoidal units [101]. Pathological hypertrophy is linked to increased production of humoral and neural mediators, hemodynamic overload, and injury to cardiomyocytes, thereby contributing to severe CVDs, including HF and arrhythmia, ultimately resulting in organ failure. Due to ACh ameliorates ER stress in ECs after H/R the obscure and complicated underlying mechanisms, there is currently no effective treatment for CH. Therefore, it is crucial to identify strategies to inhibit CH.

Adverse electrical remodeling frequently coexists with CH, as demonstrated by aberrant activities of K^+^ and L-type Ca^2+^ channels [102,103]. Overexpression of M_3_-mAChR accelerated cardiac repolarization and shortened AP duration prevented CH-induced QTc interval prolongation, by reducing AP duration, K^+^ current was subsequently increased to promote repolarization to prevent detrimental electrical remodeling [76]. This may be a new strategy for antiarrhythmic therapy in CH patients.

Ang II induces cardiomyocyte hypertrophy, resulting in various ailments, such as CH [49]. Upregulation of M_3_-mAChR expression was reported to attenuate increased expression of atrial natriuretic peptide and the β-myosin heavy chain in Ang II-induced CH [21], improve left ventricular hypertrophy and ejection fraction [104], protect against CH by downregulating expression of Ang II receptor type 1 [20], and normalize dysregulated expression of the anti-hypertrophic factor miR-133a [105]. Several studies have shown that M_3_-mAChR exerts cardioprotective effects on AngII-induced CH through a variety of different pathways.

Nevertheless, further studies are needed to promote the application of M_3_-mAChR modulators as novel therapeutic agents for CH.

### 6.3. M_3_-mAChR in Heart Failure

HF is a chronic and progressive syndrome induced by structural or functional cardiac abnormalities, leading to reduced or preserved left ventricular ejection fraction [106]. Risk factors for HF include high blood pressure, diabetes mellitus, obesity, smoking, coronary artery disease, and heredity [107]. Correction of the imbalance between sympathetic and parasympathetic nerve activity is effective for the treatment of HF [108]. Additionally, the vagal neurotransmitter ACh, which exhibits strong cardioprotective effects, is synthesized from choline as a precursor [50]. Choline can enhance vagal activity and regulate the metabolism of ketones and fatty acids.

Parasympathetic activation phosphorylation ryanodine receptor 2 (RyR2), which has an increased cardiac sarcoplasmic reticulum calcium library of effective consumption, shows that CVDs, especially one of the reasons for the occurrence of HF development, are an obstacle for RyR2 function. RyR2 Ser-2814 phosphorylation decreases due to M_3_-mAChR reducing parasympathetic nerve stimulation and the reactivity of Ca^2+^/CaMKII dependence, leading to increased systolic Ca^2+^ release, and it decreases the leakage of abnormal Ca^2+^, thus improving the Ca^2+^ cycle efficiency. Moreover, Ca^2+^ recovery cycles are good for HF. This finding provides a new research direction for the treatment of HF [109].

AF is widespread in HF. Unlike the normal heart, I_KM3_ exhibits a greater current density in AF compared to I_KACh_ and I_K4AP_. Alterations in M_3_-mAChR subtype quantities have been linked to fluctuations in K^+^ currents. Atrial remodeling in AF in the context of congestive HF could be impacted by these modifications [110].

In clinical studies, M_3_-mAChR antagonists have also been found to improve the chronic symptoms of HF, such as left ventricular end-diastolic pressure and brain natriuretic peptide levels, in chronic obstructive pulmonary disease. M_3_-mAChR may be a key target for the treatment of HF, and it is worth studying it further [111].

These findings demonstrate that M_3_-mAChR plays a significant role in the development of HF; thus, continued research is warranted to further clarify this relationship.

## 7. Future Directions

Although the research on M_3_-mAChR in CVDs is relatively clear at present, there are still many unexplored molecular mechanisms waiting to be discovered and verified. In some studies, researchers found that increased M_3_-mAChR activity in Dahl salt-sensitive rats was associated with blood pressure and that a knockout of M_3_-mAChR could reduce blood pressure and improve salt-induced hypertension, which is worth noting. The research on the role of non-myocardial cells in CVDs is relatively limited, making it easy to overlook the potential role of M_3_-mAChR in myocardial fibrosis and remodeling. Similarly, there is still a lack of specific target M_3_-mAChR inhibitors for the heart. Some studies on CVD research may disregard the effects of M_2_-mAChR, but they still cannot definitively attribute the results solely to M_3_-mAChR.

To successfully translate theoretical findings into clinical practice, M_3_-mAChR should be considered as a therapeutic target to develop more precise agonistic/inhibitory drugs and to prevent unexpected side effects by developing individualized treatment regimens. It is also important to study how M_3_-mAChR improves the prognosis of other diseases to lay a stronger theoretical foundation for further research and the investigation of M_3_-mAChR’s role in CVDs; to develop more powerful therapeutic drugs; and to formulate practical treatment strategies.

## 8. Conclusions and Perspectives

M_3_-mAChR is widely distributed throughout the body. The function of M_3_-mAChR in the heart and its associated pathological consequences are discussed in this article. M_3_-mAChR regulates the expression of numerous proteins and regulatory factors to maintain the cardiovascular system in a state of homeostasis. It is present in cardiomyocytes, fibroblasts, and ECs. Numerous aspects of M_3_-mAChR’s impact on CVDs have been investigated, including in vivo studies in animal models and in vitro studies at the cellular and protein levels. The best marker for the identification and management of CVDs is M_3_-mAChR. In several cardiovascular conditions, such as CH, I/R injury, HF, and other conditions, it can be highly protective.

## Figures and Tables

**Figure 1 ijms-25-07560-f001:**
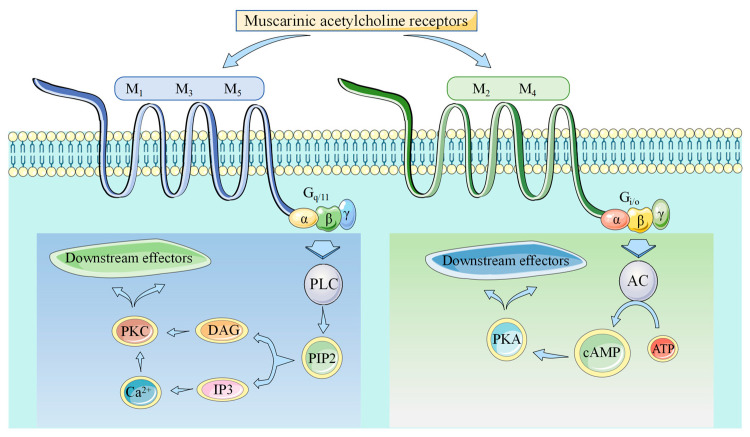
Subclassification of muscarinic acetylcholine receptors and the downstream effectors. AC, adenylyl cyclase; ATP, adenosine triphosphate; cAMP, cyclic adenosine monophosphate; DAG, diacylglycerol; IP3, inositol trisphosphate; M_1_, muscarinic acetylcholine receptor M_1_; M_2_, muscarinic acetylcholine receptor M_2_; M_3_: muscarinic acetylcholine receptor M_3_; M_4_, muscarinic acetylcholine receptor M_4_; M_5_, muscarinic acetylcholine receptor M_5_; PIP2, phosphatidylinositol 4,5-bisphosphate; PKA, protein kinase A; PKC, protein kinase C; PLC, phospholipase C.

**Figure 2 ijms-25-07560-f002:**
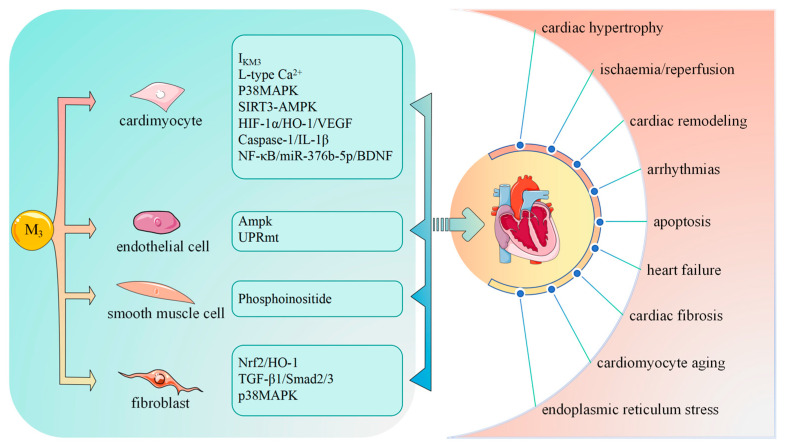
M_3_-mAChR functions in the cardiac system through different cardiomyocytes. M_3_-mAChR decreases Ca^2+^ overload, aiding in cytoprotection, enhancing cardiac function, activating delayed rectifier potassium current I_KM3_ to regulate heart rate and repolarization, and interacting with the gap junction channel, Cx43, which may help coordinate repolarization rate in cardiomyocytes. M_3_-mAChR may utilize several signaling pathways to generate various protective effects against cardiovascular diseases. AMPK, AMP-activated protein kinase; L-type Ca^2+^, L-type calcium; UPRmt, mitochondrial unfolded protein response; p38MAPK, p38 mitogen-activated protein kinase; NF-κB, nuclear factor-kappa B; miRNA, microRNA; BDNF, brain-derived neurophic factor; IL-1β, interleukin-1β; TGF-β1, transforming growth factor beta; HIF-1α, hypoxia-inducible factor 1; HO-1, heme oxygenase-1; VEGF, vascular endothelial growth factor; mTOR, mammalian target of the rapamycin; HSF1, heat shock transcription factor 1; SIRT3-AMPK, sirtuin 3/AMP-activated protein kinase; Nrf2, nuclear factor erythroid 2-related factor 2.

**Table 1 ijms-25-07560-t001:** Main functions and roles of M3-mAChR in cardiovascular diseases.

Cell Types	Main Mechanism	Functional Outcome	Cardiovascular Significance
cardiomyocytes	potassium currents and repolarization [76]	adverse electrical remodeling	cardiac hypertrophy
	SIRT3-AMPK [49]	metabolic dysfunction in the heart	
	p38 MAPK [21]	attenuated the increment cell size	
	L-type Ca^2+^ currents [72]	AP shortening	arrhythmia
	Notch1/HSF1 [56]	impedes oxidative damage and cardiomyocyte apoptosis	early stage heart failure
	HIF-1α, HO-1, VEGF [57]	cytoprotection	apoptotic
	NF-κB, miR-376b-5p, BDNF [3]	cardioprotection	ischemia-induced cardiac injury
	IL-1β, Caspase-1 [2]	cardiomyocyte aging	age-related cardiac impairment
endothelial cells	AMPK [55,70]	ER stress and apoptosis [55], endothelium damage O-glycosylation, and apoptosis [70]	ischemia/reperfusion injury [55]
	VDAC1/glucose-regulated protein 75/inositol 1,4,5-trisphosphate receptor 1 complex and mitofusin 2 [80]	ER-mitochondria Ca^2+^ cross talk	reperfusion injury, endothelial protection
	mtROS, mitonuclear protein imbalance [81]	unfolded protein response	hypoxia/reoxygenation, endothelial cell damage
	maintaining PTP1B activity, keeping the adherens junction proteins dephosphorylation [82]		preserves the endothelial barrier function
smooth muscle cells	Phosphoinositide [87]		contraction in smooth muscle
fibroblasts	Nrf2, HO-1 [95]	inflammation	DOX-induced cardiotoxicity
	TGF-β1, Smad2/3, p38MAPK [19]	interstitial fibrosis, collagen I and III production	fibrosis

AMPK, AMP-activated protein kinase; AP, action potential; BDNF, brain-derived neurophic factor; ER, endoplasmic reticulum; HSF1, heat shock transcription factor 1; HO-1, heme oxygenase-1; IL-1β, interleukin-1β; L-type Ca^2+^, L-type calcium; mtROS, mitochondrial reactive oxygen species; NF-κB, nuclear factor-kappa B; Nrf2, nuclear factor erythroid 2-related factor 2; p38MAPK, p38 mitogen-activated protein kinase; SIRT3-AMPK, sirtuin 3/AMP-activated protein kinase; TGF-β1, trans-forming growth factor β; VDAC1, voltage-dependent anion channel-1; VEGF, vascular endothelial growth factor.

## Data Availability

Data are contained within the article.

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
