# Peer review of "The Role of Muscarinic Acetylcholine Receptor M3 in Cardiovascular Diseases"

_ijms, 2024, doi:10.3390/ijms25147560_

Round 1

Reviewer 1 Report

Comments and Suggestions for Authors

In this Review article, Authors sought to report literature data regarding the physiological and physio-pathological role of muscarinic acetylcholine receptor M3 in cardiovascular diseases. The topic is the of physiological interest, and the review is well structured, but the presentation suffers from shortcomings.

Criticism and suggestion are indicated below:

- Abstract section should be improved. Authors should report other information about the role of muscarinic acetylcholine receptor M3 in CVD for the readability of review content.

-  The finds presented in the manuscript are mainly summarized. The authors often did not actively review the evidence available, but just presented it. The authors should strive to do so more thoroughly in particular in ischemia/reperfusion injury, cardiac hypertrophy, and heart failure, as mentioned in the abstract section.

- Numerous literature data reporting a diagnostic and therapeutic role of muscarinic acetylcholine receptor M3, as mentioned by authors, but this important point suffers shortcomings and was not discussed in depth. Please, include further studies, even recent ones, to improve this aspect.

- A synoptic table recapitulating the main functions of muscarinic acetylcholine receptor M3 in CVD could be appropriate and may help the readers.

- Several typos are present in the text.

Comments on the Quality of English Language

Moderate editing of English language required

Author Response

Comments 1: Abstract section should be improved. Authors should report other information about the role of muscarinic acetylcholine receptor M3 in CVD for the readability of review content.

Response 1: Thank you for your suggestion. We have made some modifications. As shown on page 2, we added the protective effect of muscarinic acetylcholine receptor M3 in CVD, and further enriched the content.

Comments 2:  The finds presented in the manuscript are mainly summarized. The authors often did not actively review the evidence available, but just presented it. The authors should strive to do so more thoroughly in particular in ischemia/reperfusion injury, cardiac hypertrophy, and heart failure, as mentioned in the abstract section.

Response 2: Thank you for pointing this out, we agree with this and have revised it with your comments. We have marked the modified areas, at Lines 16, 22, and 28 on page 9, lines 17, and 27 on page 10, and lines 3 on page 11. We discussed and further expanded the existing evidence on ischemia/reperfusion injury, myocardial hypertrophy, and heart failure.

Comments 3: Numerous literature data reporting a diagnostic and therapeutic role of muscarinic acetylcholine receptor M3, as mentioned by authors, but this important point suffers shortcomings and was not discussed in depth. Please, include further studies, even recent ones, to improve this aspect.

Response 3: Thank you for pointing this out, we agree and have amended it based on your comments. Lines 16, 22, and 28 on page 9, lines 17 on page 10, and lines 3 on page 11 are the updated sections that we have indicated. We have deliberated on and expanded upon the existing evidence concerning ischemia/reperfusion injury, myocardial hypertrophy, and heart failure.

Comments 4: A synoptic table recapitulating the main functions of muscarinic acetylcholine receptor M3 in CVD could be appropriate and may help the readers.

Response 4: Thank you for your advice. We quite agree with your idea and have revised it according to your comments. In the manuscript page 8 of the 10th column added provides an overview of the role of muscarinic acetylcholine receptor M3 in CVD form, make it easier for the reader to understand the article content.

Comments 5: Several typos are present in the text.

Response 5: We sincerely apologize for our carelessness. We have diligently reviewed and rectified the errors.

Comments 6: Moderate editing of English language required

Response 6:  We have sought to polish company for polishing, polishing company name for internationalscienceediting

Reviewer 2 Report

Comments and Suggestions for Authors

M3 muscarinic acetylcholine receptor is one of the muscarinic receptor subtypes that mediates the actions of acetylcholine, a crucial neurotransmitter in the parasympathetic nervous system. Many physiological processes, including as the control of smooth muscle contraction, exocrine gland secretion, and heart rate regulation, are mediated by this receptor. It has a multifaceted and intricate role in cardiovascular disease. In order to examine the expression patterns, roles, and underlying mechanisms of action of M3-mAChR in cardiovascular disorders, the authors proposed this review. The authors provide a summary of the several cardioprotective effects that result from M3-mAChR activation in cardiovascular illnesses, with a focus on heart failure, cardiac hypertrophy, and ischemia/reperfusion injury.

I propose elaborating on a few methodological issue: is this a narrative review? For this research, which databases—and how many—did the authors use? What search terms did they use? Were MeSH terms used?

Moreover, I suggest to the authors to consider the relationship between frailty and heart disease (10.1186/s12877-021-02304-9). 

Author Response

Comments 1: I propose elaborating on a few methodological issue: is this a narrative review? For this research, which databases—and how many—did the authors use? What search terms did they use? Were MeSH terms used?

Response 1: We thank you very much for your professional comments on our articles, as you are concerned, this is a narrative review, we use PubMed, in which the use of keywords is: muscarinic acetylcholine receptor M3, cardiovascular diseases, endothelial cell, cardiomyocyte, smooth muscle cell, fibroblast, ischemia/reperfusion injury, cardiac hypertrophy, heart failure. MeSH terms were used to assist in searching the article data.

Comments 2: Moreover, I suggest to the authors to consider the relationship between frailty and heart disease (10.1186/s12877-021-02304-9). 

Response 2: We appreciated your suggestion and followed it when reading this text, which helped us understand the association between cardiac disease and frailty (S12877 10.1186-021-02304-9). Subsequently, we investigated the connection between the muscarinic acetylcholine receptor M3 and frailty. Unfortunately, no publications are linking the muscarinic acetylcholine receptor M3 to frailty, but this provides some insights for future research.

Reviewer 3 Report

Comments and Suggestions for Authors

This is a very intersting article about the  role of muscarinic receptors in cardiovascular diseases. Overall, the article is well written, and summarizes very well the scientific literature in the field. 
I have some minor comments, that should be taken into account for the revision:

- the title of the article focuses on cardiovascular diseases; however, most of the article is focused on the biomolecular mechanisms/physiology of muscarinic recepts. The authors should increase the proportion of the manuscript dedicated to the actual involvement of M3R in CV disorders. 

- the article fails to present a series of recent discoveries regarding M3Rs and their role in cardiac diseases (e.g its association with salt sensitivity and subsequently hypertension, or  their involvement in inhibition of myocardial fibrosis

- the text has different sizes.

- the authors should specify if the used images were done by them (case in which I wouldlike to see the software used), third parties (which would be added to the acknowledgment), or other sources (which should be cited)

Author Response

Comments 1: - the title of the article focuses on cardiovascular diseases; however, most of the article is focused on the biomolecular mechanisms/physiology of muscarinic recepts. The authors should increase the proportion of the manuscript dedicated to the actual involvement of M3R in CV disorders. 

Response 1: Thank you very much for your professional advice. We agree with this comment. We have made additions and modifications on lines 314,322,328, and351 on page 9, and lines 361, and 375 on page 10.

Comments 2: - the article fails to present a series of recent discoveries regarding M3Rs and their role in cardiac diseases (e.g its association with salt sensitivity and subsequently hypertension, or  their involvement in inhibition of myocardial fibrosis

Response 2: Thank you for your suggestion, and we have made a supplementary explanation for your suggestion in the Future Directions section on line 405 of page 11.

Comments 3: - the text has different sizes.

Response 3: We sincerely thank you for your careful reading, and in our resubmitted manuscript, the font has been set uniformly.

Comments 4: - the authors should specify if the used images were done by them (case in which I wouldlike to see the software used), third parties (which would be added to the acknowledgment), or other sources (which should be cited)

Response 4: We sincerely thank you for your valuable suggestions and we illustrate the source of picture 1 on page 2, line 55 of the manuscript. Figures 2 and 3 are drawn with the help of PowerPoint software.

Reviewer 4 Report

Comments and Suggestions for Authors

The article entitled "The role of muscarinic acetylcholine receptor M3 in cardiovascular diseases" is very interesting, about a new concept with many future persectives. However, the information need to be a little bit adjust in order to be more understable.

1. First of all, the bibliography is not written correctly. Please read how the bibliography needs to be written for the MDPI (and in text - you need to put numbers).

2. The abstract is too short. The authors need to write in the abstract the importance of this subject and what they concluded.

3. Are the figure 1, 2  original? If not, you need to ask permission for them. If they are original, please mention this.

4. Row 62 - you have a dot instead of coma.

5. Subtitles are needed to be written in extense, not with abbreviation. 

6. How was the research from the literature make? The authors should insert a paragraph about the materials and methods, how the research was make, the number of the results (articles), how did they include and exclude the articles etc.

7. Pharagraphs 6.1 and 6.1 have different font and size. Please revise.

8. The authors should make a different section for the future perspectives and include more information.

9. Why is this review important? The authors should mention something about the medication, if there are some future perspective about medication or other studies and if it can influence somehow the evolution of some cardiovascular diseases. Also, the author's personal opinion is important.

10. The authors should present a short and clear conclusion. Please revise the conclusion.

11. The authors wrote in row 420, 421 that the mechanism remain unclear. However, they presented in the manuscript some mechanism. Please comment.

Comments on the Quality of English Language

Just minor English editing.

Author Response

Response 1: Thank you very much for your careful reading. The format of the references has been modified.

Comments 2: The abstract is too short. The authors need to write in the abstract the importance of this subject and what they concluded.

Response 2: We are very grateful for your guidance. A brief introduction outlining the significance of our story and conclusions has been added to the summary on page 2, which we have updated and added to.

Comments 3: Are the figure 1, 2 original? If not, you need to ask permission for them. If they are original, please mention this.

Response 3: Thank you very much for reading our article carefully. Figure 1 was downloaded from the PDB database, and Figure 2 was drawn by us with the help of PowerPoint software

Comments 4: Row 62 - you have a dot instead of coma.

Response 4: We feel very sorry for our careless mistake, in the resubmitted manuscript, error has been modified, thank you for your correct.

Comments 5: Subtitles are needed to be written in extense, not with abbreviation. 

Response 5: Thank you for yoursuggestion interest. We have made corrections to the abbreviation of the title and added annotations throughout the manuscript. This includes corrections on page 2 line 34, page 6 lines 7, 12, and 21, page 7 line 5, page 8 line 7, page 9 line 8, page 10 line 8, and 31.

Comments 6: How was the research from the literature make? The authors should insert a paragraph about the materials and methods, how the research was make, the number of the results (articles), how did they include and exclude the articles etc.

Response 6: We appreciate you taking the time to carefully read our essay, and we appreciate your insightful comments. After searching PubMed for publications, we examined the study findings about muscarinic acetylcholine receptor M3 and cardiovascular disease in the pertinent literature and reviewed the literature.

Comments 7: Pharagraphs 6.1 and 6.1 have different font and size. Please revise.

Response 7: We are sorry for our carelessness, thanks you for your reminder, we have modified this paragraph in the new manuscript.

Comments 8: The authors should make a different section for the future perspectives and include more information.

Response 8: We appreciate your thorough evaluation of our work, and we have included your suggestions. A future vision is now included in the manuscript's highlighted section on page 11.

Comments 9: Why is this review important? The authors should mention something about the medication, if there are some future perspective about medication or other studies and if it can influence somehow the evolution of some cardiovascular diseases. Also, the author's personal opinion is important.

Response 9: Thank you for your professional advice, according to our review, the muscarinic acetylcholine receptor M3 is a prime target for cardiovascular disease treatment and may be linked to the onset and progression of cardiovascular disorders. Drugs that can either activate or inhibit the cardiac muscarinic acetylcholine receptor M3 are a significant area of inquiry that merits further consideration and investigation.

Comments 10: The authors should present a short and clear conclusion. Please revise the conclusion.

Response 10: Thank you very much for your advice, we have revised according to your suggestions this part, in the eleventh page highlight section of the manuscript.

Comments 11: The authors wrote in row 420, 421 that the mechanism remain unclear. However, they presented in the manuscript some mechanism. Please comment.

Response 11: Thank you very much for your valuable feedback. We have made adjustments to the relevant section of the document since our language description in this section is unsuitable. To improve the theoretical foundation for the role of muscarinic acetylcholine receptor M3 in the pathophysiology of cardiovascular disorders, researchers still need to delve deeper into the precise mechanism of this receptor in the heart.

Round 2

Reviewer 4 Report

Comments and Suggestions for Authors

The authors made most of the necessary adjustment. However, some minor revision are still needed to be made:

1. The last paragraph should be just the conclusion, the future perspectives are needed to be discussed before.

2. Do you have the agreement from the database to use figure 1? You should have a write consent to use it. And also in the description of the figure please mention from where it is.

3. The authors should also add a paragraph in the manuscript about how the research of the articles was performed (how many did they find in the literature, the exclusion criteria of some articles, etc. as I mentioned before)

Author Response

Comments 1: The last paragraph should be just the conclusion, the future perspectives are needed to be discussed before.

Response 1: Thank you for your valuable professional insights. We have adopted your suggestions and changed the position of the contents of the two parts of future directions, conclusions, and perspectives, made necessary modifications on page 10, and highlighted them in red.

Comments 2: Do you have the agreement from the database to use figure 1? You should have a write consent to use it. And also in the description of the figure please mention from where it is.

Response 2: Thank you very much for your careful review of our paper. We are very sorry that we have not yet obtained written permission for Figure 1. Therefore, we have removed Figure 1 from the article and made modifications on page 2, line 59, and page 6, line 276.

Comments 3: The authors should also add a paragraph in the manuscript about how the research of the articles was performed (how many did they find in the literature, the exclusion criteria of some articles, etc. as I mentioned before)

Response 3: Thank you for your professional advice, which we have accepted. On page 11 of the manuscript, line 421, we have added a paragraph to describe how the research of our article was conducted, such as how to find articles and exclusion criteria for articles.